# Differential behaviour of a risk score for emergency hospital admission by demographics in Scotland—A retrospective study

**Ioanna Thoma**[1,2], **Simon Rogers**[3], **Jillian Ireland**[4], **Rachel Porteous**[4], **Katie Borland**[4], **Catalina A. Vallejos**[1,2]*, **Louis J. M. Aslett**[1,5]*, **James Liley**[1,5]*

**1** Alan Turing Institute, London, United Kingdom, **2** MRC Human Genetics Unit, Institute of Genetics and Cancer, University of Edinburgh, United Kingdom, **3** NHS National Services Scotland, United Kingdom, **4** Public Health Scotland (PHS), Edinburgh, Scotland, **5** Department of Mathematical Sciences, Durham University, United Kingdom

* catalina.vallejos@ed.ac.uk (CAV); louis.aslett@durham.ac.uk (LJMA); james.liley@durham.ac.uk (JL)

**Data Availability Statement:** Raw data for this project are patient-level electronic health records for the Scottish population, which have been

## Abstract

The Scottish Patients at Risk of Re-Admission and Admission (SPARRA) score predicts individual risk of emergency hospital admission for approximately 80% of the Scottish population. It was developed using routinely collected electronic health records, and is used by primary care practitioners to inform anticipatory care, particularly for individuals with high healthcare needs. We comprehensively assess the SPARRA score across population subgroups defined by age, sex, ethnicity, socioeconomic deprivation, and geographic location. For these subgroups, we consider differences in overall performance, score distribution, and false positive and negative rates, using causal methods to identify effects mediated through age, sex, and deprivation. We show that the score is well-calibrated across subgroups, but that rates of false positives and negatives vary widely, mediated by various causes including variability in demographic characteristics, admission reasons, and potentially differential data availability. Our work assists practitioners in the application and interpretation of the SPARRA score in population subgroups.

## Author summary

Prediction models can potentially improve biomedical outcomes and healthcare. However, they might exhibit variation in performance across demographic groups, even if well-calibrated overall. Evaluation of predictive accuracy in population subgroups, considered alongside data collection processes, can help ensure that health disparities are not increased for those already underserved. We examined predictive performance of the 'SPARRA' predictive score across a range of demographic groups using metrics of statistical fairness such as false positive and negative rates. We identified differential accuracy associated with age, sex, socioeconomic deprivation, ethnicity and residence location (mainland/island; urban/rural). We explored patterns of prediction errors across medical

anonymised for confidentiality ahead of any analysis being undertaken, and are not publically available. Enquiries about access to this data may be directed to phs.edris@phs.scot. The project was covered under National Safe Haven Generic Ethical Approval (favourable ethical opinion from the East of Scotland NHS Research Ethics Service). This study was conducted in accordance with UK data governance regulations and the use of patient-level EHR was approved by the Public Benefit and Privacy Panel (PBPP) for Health and Social Care (approval evidenced in application outcome minutes for 2018/19 at https://www. informationgovernance.scot.nhs.uk/pbpphsc/ application-outcomes/). Data access was also approved by the PHS National Safe Haven, through the electronic Data Research and Innovation Service (eDRIS). All studies have been conducted in accordance with information governance standards; data had no patient identifiers available to the researchers. Due to the confidential nature of the data, all analysis took place on a remote 'data safe haven', without access to internet, software updates or unpublished software. Information Governance training was required for all researchers accessing the analysis environment. To avoid the risk of accidental disclosure of sensitive information, an independent team carried out statistical disclosure control checks on all data exports, including the outputs presented in this manuscript. However, the summary data required to draw figures included in our manuscript is publically available from https://github.com/Public-Health-Scotland/sparra-performance-analysis, as is all code to regenerate figures.

**Funding:** This study was supported by The Alan Turing Institute, PHS, the MRC Human Genetics Unit at the University of Edinburgh, Durham University and Health Data Research UK. This project is supported by the Health Foundation, an independent charity committed to bringing about better health and health care for people in the UK. IT, JL, CAV, LJMA were supported by the Wave 1 of The UKRI Strategic Priorities Fund under the EPSRC Grant EP/T001569/1 and EPSRC Grant EP/W006022/1, particularly the "Health and Medical Sciences" theme within those grants and The Alan Turing Institute. IT, JL, CAV, LJMA were also supported by Health Data Research UK, an initiative funded by UKRI, the Department of Health and Social Care (England), the devolved administrations, and leading medical charities. LJMA was partially supported by a Health Programme Fellowship at The Alan Turing Institute. The funders had no role in the study design, data collection, analysis, decision to publish, or preparation of the manuscript.

causes of emergency admission, finding that, to variable degrees across subgroups, cardiac and respiratory admissions are more likely to be correctly predicted from electronic health records. This work provides an atlas of performance measures for SPARRA and partly explains how between-group performance differences arise. Our findings indicate that the precision with which the SPARRA score predicts emergency hospital admissions differs between population subgroups. Differences are largely driven by performance variation across age and sex and predictability of different causes of admission. Awareness of these differences is important when making decisions based on the SPARRA score.

## Introduction

The UK's healthcare system has been repeatedly reported to be under significant pressure due to increasing workload, hospital demand, and the resulting strain on workforce and resources [1, 2]. The COVID-19 pandemic and its aftermath have intensified the challenges faced by human resources in primary care [3]. Consequently, proactive interventions to prevent individuals from experiencing abrupt breakdowns in health have been highlighted as a key priority for modern medical practice [4]. In particular, emergency hospital admissions (EAs), in which urgent in-hospital care is needed for an individual, are a potential target for primary care intervention, as some of these events can be averted through appropriate anticipatory care [5–7]. However, since total primary care capacity is limited, it is important to optimise the allocation of existing resources [8]. To this end, individual-level prediction of future EAs risk can shape decision-making by helping to identify individuals who may benefit the most from anticipatory primary care intervention [9–14].

SPARRA (Scottish Patients At Risk of Readmission and Admission) is a risk prediction score calculated by Public Health Scotland (PHS) using routinely collected Scotland-wide electronic health records [15] to estimate, at an individual level, the probability of having an emergency in-patient hospital admission in the subsequent year. SPARRA scores range from 1 to 99, with higher scores indicating a higher estimated probability of admission. For a lay overview, see [16]. To date, three versions of the SPARRA score have been deployed nationally in Scotland, the first one dating back to 2006. SPARRA version 3 (referred to as SPARRAv3) [17] has been in use since 2012. Each month, PHS calculates SPARRAv3 scores for around 80% of Scottish population. A fourth version (SPARRAv4) was recently developed [15] and is expected to be deployed by 2024 on a national scale. SPARRA scores aim to support General Practitioners (GPs) as they plan anticipatory interventions for the patients under their care. For example, GPs have previously been incentivised to review patients with SPARRA scores in the range 40–60, with the intention of identifying patients not already known to be at high risk of emergency admission. At an aggregate level, the scores can also be used to e.g. predict future hospital demand.

Our objective is to study the differential behaviour and accuracy of the SPARRA score across population subgroups (such as urban or rural populations, and residents of more or less deprived geographic areas). We expect that the behaviour of the score may vary across such subgroups due to variable demographic characteristics (age, sex, and deprivation), and potentially due to differential data availability and access to healthcare [18]. A major motivation for our work was to ensure that the SPARRA score did not worsen equity in healthcare provision. We aimed to partly assess this by analysing several conceptions by which the score could in itself be considered to be making equitable predictions. Even if the SPARRA score very accurately predicts EA risk in all groups, between-group differences in demographic characteristics

**Competing interests:** The authors have declared that no competing interests exist.

may mean that decisions made on the basis of the SPARRA score may have different consequences across different groups, in that if practitioners take action on all patients for whom the SPARRA score exceeds a given threshold, the rates of false positives and false negatives may vary between groups. We consider that without awareness of these differential consequences, primary care practitioners may be unreasonably dissuaded from using SPARRA scores, potentially introducing or exacerbating existing health inequalities (in which Scotland ranks poorly compared to western and central Europe [19, 20]). Hence, it is critical to scrutinise its behaviour across a range of demographic groups, which can also help better understand the epidemiology behind EAs in Scotland and advance towards a more equitable and healthier future.

In this article, we present a retrospective comprehensive evaluation of the performance of SPARRA across a range of groupings defined by age, sex, deprivation, ethnicity, and geographic location (rural versus urban; island versus mainland residency). Our results and code are publicly available on Github and the `SPARRAfairness` R package. We also provide an online Shiny application for interactive exploration of our results. We intend it to be usable by primary care practitioners and the public to enable informed decision-making and enhance the interpretation of SPARRA scores.

## Materials and methods

### Data

Our primary analysis is based on the third version of the score (SPARRAv3), deployed at a national level in Scotland since 2012 [17] (see S1 Appendix, Section 2, for details).

We repurposed the same retrospective data and inclusion/exclusion criteria used by [15], focusing on a single prediction time cutoff (May 1st, 2016). The data comprise every acute hospital record (EAs, elective admissions, day cases, outpatient attendances, A&E attendances, and records of long-term conditions) and community prescribing activity within the National Health Service (NHS) Scotland up to three years before the time cutoff. We also use age, sex, and mortality records, and long-term condition (LTC) records dating back to January 1981 (when records began) to capture any long-term conditions an individual has acquired throughout their life.

For this analysis, the data was extended to include information that is not currently used as input for SPARRA. First, self-reported ethnicity information was obtained by PHS through cross-reference with a range of datasets, including the COVID-19 vaccination programme (see S1 Appendix, Section 10). Moreover, PHS used postcode information to derive urban/rural and mainland/island residence status indicators (considered dichotomously as whether an individual lives on the mainland or on any Scottish island). The separation between urban and rural postcodes was based on the Scottish government's 2016 classification system [21].

### Definition of demographic groups

We consider various aspects of the performance of SPARRAv3 over demographically-defined cohorts of patients defined by age, sex (female/male as per current Scottish community health index number), SIMD quintiles (with lower quintiles indicating higher deprivation), ethnicity and indicators for urban/rural and mainland/island residence status. Hereafter, we refer to these as *grouping variables*. For each grouping variable, missing values were excluded when defining the associated groups (rates of missingness are shown in Table 1). For ease of interpretation, we considered only a two-group comparison for age and SIMD. Rather than dichotomising the entire population, we chose groups to highlight the differences between the extremes of these variables (over 65 and under 25 for age; most and least deprived quintiles for

**Table 1. Descriptive statistics stratified across groups (in columns).** N was rounded to the nearest thousand. All numbers are percentages unless indicated otherwise. *Italicised* groups are not specifically analysed. 'Most': in the most deprived quintile; 'Least': in the least deprived quintile; 'Miss.': missing; 'Main./Isl.': mainland/island. There were no missing values for age, sex, or deprivation.

| | *All* | Sex | | Age | | | Deprivation | | | Ethnicity | | | Urban/rural | | | Main./Isl. | |
|---|---|---|---|---|---|---|---|---|---|---|---|---|---|---|---|---|---|
| | | M | F | ≥ 65 | ≤ 25 | *26–64* | Most | Least | *Other* | W | NW | *Miss.* | U | R | *Miss.* | ML | IL |
| **N (thousand)** | *4286* | 1946 | 2340 | 884 | 1100 | *2302* | 923 | 803 | *2560* | 2862 | 933 | *491* | 3571 | 713 | *3* | 4204 | 80 |
| **Sex and age** | | | | | | | | | | | | | | | | | |
| Male | *45.4* | 100 | 0 | 44.1 | 47.5 | *44.9* | 46.2 | 44.8 | *45.3* | 44.3 | 47.9 | *47.1* | 45.2 | 46.2 | *50.9* | 45.4 | 46.7 |
| Age (years, mean) | *43.3* | 42.7 | 43.7 | 75.5 | 12.1 | *45.8* | 40.4 | 44.1 | *44* | 41.4 | 39.3 | *61.8* | 42.7 | 46 | *42* | 43.2 | 47.3 |
| Age (years, SD) | *23.8* | 23.9 | 23.7 | 7.22 | 7.51 | *11.7* | 23.2 | 24.1 | *23.7* | 22.6 | 22.1 | *31.7* | 23.7 | 23.9 | *22.2* | 23.7 | 24 |
| **Deprivation** | | | | | | | | | | | | | | | | | |
| Most deprived | *21.5* | 21.9 | 21.2 | 17.1 | 23.8 | *22.2* | 100 | 0 | *0* | 22.7 | 18.2 | *21.3* | 24.9 | 4.49 | *22.5* | 21.9 | 2.13 |
| Least deprived | *18.7* | 18.5 | 18.9 | 20.3 | 18.9 | *18.1* | 0 | 100 | *0* | 18.1 | 20.6 | *18.7* | 20.7 | 9.09 | *12.4* | 19 | 2.23 |
| **Ethnicity** | | | | | | | | | | | | | | | | | |
| White (W) | *66.8* | 65.1 | 68.1 | 54.5 | 69.7 | *70.1* | 70.3 | 64.6 | *66.2* | 100 | 0 | *0* | 67.2 | 64.4 | *62.2* | 66.9 | 58.3 |
| Nonwhite (NW) | *21.8* | 23 | 20.8 | 13.9 | 24.9 | *23.3* | 18.3 | 23.9 | *22.3* | 0 | 100 | *0* | 21.3 | 24 | *27.1* | 21.6 | 29 |
| **Postcode** | | | | | | | | | | | | | | | | | |
| Urban (U) | *83.3* | 83 | 83.6 | 80.3 | 85 | *83.7* | 96.5 | 91.9 | *75.9* | 83.9 | 81.6 | *83.2* | 100 | 0 | *0* | 84.5 | 26 |
| Rural (R) | *16.6* | 16.9 | 16.4 | 19.7 | 15 | *16.2* | 3.47 | 8.07 | *24* | 16 | 18.3 | *16.8* | 0 | 100 | *0* | 15.5 | 74 |
| Mainland (ML) | *98.1* | 98 | 98.1 | 97.6 | 98.4 | *98.1* | 99.7 | 99.7 | *96.9* | 98.3 | 97.4 | *97.9* | 99.4 | 91.7 | *0* | 100 | 0 |
| Island (IL) | *1.86* | 1.91 | 1.82 | 2.38 | 1.57 | *1.8* | 0.184 | 0.222 | *2.98* | 1.63 | 2.48 | *2.06* | 0.58 | 8.29 | *0* | 0 | 100 |

SIMD, derived from the top and bottom two deciles). Ethnicity data was also aggregated into two groups (white and non-white) due to the small sample sizes observed for some non-white groups.

Our choice of grouping variables was restricted to those identifiable from data held by PHS, whilst capturing commonly-identified sources of inequality. In particular, we considered groups defined by urban/rural postcodes and by mainland/island postcodes given the potential variation across these groups in environmental and socioeconomic factors (such as employment opportunities) and in access to healthcare [22, 23]. We considered ethnicity as a grouping variable due to its association with differential access to healthcare, socioeconomic and environmental factors, and cultural practices [24–26]. None of these variables are used as inputs for the SPARRA score. Finally, we included age, sex and deprivation (SIMD), all of which are used as inputs for SPARRA. Deprivation was considered as growing evidence suggests that health inequalities among Scotland's most and least deprived areas are substantial [27]. Age and sex were selected as risk factors that are known to influence EA susceptibility, disease patterns and outcomes.

## Metrics

To deliver a comprehensive evaluation of SPARRA and its performance across different groups, we applied a variety of metrics recently employed across the ML literature, generally in the context of between-group fairness [28–30]. These comprise measures of predictive discrimination and calibration as well as different types of error rates (see Table 2 and Supplementary Note S1 Appendix for more detailed definitions).

We will use $Y \in \{0, 1\}$ to denote the observed outcome for an individual, where $Y = 1$ means an EA or death occurs within one year from the prediction time cutoff. We use $G$ to denote the group under consideration to which the individual belongs. We denote the SPARRAv3 score for the individual as $\hat{Y}$. We consider a hypothetical decision rule that binarises

**Table 2. Metrics used to assess score and a brief interpretation, including the quantity we aim to estimate.** We compare estimated probabilities across two groups, $G = g$ and $G = g'$, and a particular score cutoff $c$.

| Category | Metric | Interpretation |
|---|---|---|
| Demographic parity | Cumulative distribution function (CDF) $P(\hat{Y} < c | G = g)$ | The proportion of people in a given group that have a SPARRA score less than the cutoff $c$ |
| | Counterfactual CDF | The proportion of people for whom the SPARRA score is less than the cutoff $c$ in a hypothetical group that consists of members of $g'$ whose medical history resembles the corresponding distribution amongst those in group $g$. |
| Discrimination and calibration | ROC curve/ AUROC $P(\hat{Y}_1 > \hat{Y}_2 | G = g, Y_1 > Y_2)$ for samples $Y_1$ and $Y_2$ | Within a group $g$, the probability that an individual who experienced the event has a higher score than someone who did not experience the event |
| | Within-group calibration $P(Y = 1 | G = g, \hat{Y} = c)$ | The proportion of people who experienced the event amongst those in group $g$ whose predicted score is $c$ |
| False negatives | False omission rate (FOR) $P(Y = 1 | G = g, \hat{Y} < c)$ | Amongst individuals in group $g$ with a score less than a cutoff $c$, the proportion that experienced the event |
| | Adjusted FOR | Amongst individuals in a group with a given fixed age, sex and SIMD decile (technically, a fixed distribution) and a score less than the cutoff, the proportion that experienced the event |
| False positives | False discovery rate (FDR) $P(Y = 0 | G = g, \hat{Y} \geq c)$ | Amongst individuals in a group $g$ with a score greater than the cutoff $c$, the proportion that did not experience the event |
| | Adjusted FDR | Amongst individuals in a group $g$ with a given fixed age, sex and SIMD decile (technically, a fixed distribution) and with score greater than the cutoff $c$, the proportion that did not experience the event |

those predictions such that the decision is to predict $Y = 1$ for an individual if and only if $\hat{Y} > c$ and consider the consequences of varying $c$ across the interval [1, 99]%.

We first assess 'demographic parity' by comparing the cumulative distribution of scores $\hat{Y}$ in the grouped populations, i.e., $P(\hat{Y} < c | G = g)$. We then compare the distribution of scores under a 'counterfactual' setting in which we substitute a grouping value for another (e.g. rural instead of urban), while the underlying distribution of hospital activity data and prescriptions is held constant [31, 32]. This isolates the effects of the grouping variable on the score to only those mediated through either the direct effect of the group or through differences in distributions of age, sex and SIMD.

We evaluated group-level predictive performance using receiver-operator characteristic curve (ROC) and calibration curves. The area under the ROC (AUROC) quantifies the ability of the model to rank individuals accurately based on predicted risk; we thus evaluate the model's ability to discriminate between predicted admissions versus non-admissions for all groups under consideration. Note that predicted risk can inaccurately represent observed risk even if the model attains good discrimination [33]. Instead, calibration assesses the agreement between the observed and predicted number of events [34]. A risk score that is well-calibrated for group $g$ should have $P(Y = 1 | G = g, \hat{Y} = c) \approx c$ for most $c$; that is, amongst individuals with $\hat{Y} = c$, a proportion of around $c$ have $Y = 1$. We assessed calibration visually by directly plotting calibration curves (sometimes called reliability diagrams [35]) in each group.

We then assess the rates of false-negative and false-positive errors [36] by group based on the hypothetical decision rule. A false negative means a setting in which the SPARRA score predicted that an individual would not have an event (i.e., $\hat{Y} < c$, for a given cutoff value $c$), but that individual did have an event (i.e. $Y = 1$ is observed). Correspondingly, a false positive means a setting for which $\hat{Y} \geq c$ but $Y = 0$. Within a group $g$, the probability of observing a false negative is $P(Y = 1 | G = g, \hat{Y} < c)$, and the probability of observing a false positive is $P(Y = 0 | G = g, \hat{Y} \geq c)$, which we term false omission rate (FOR) and false discovery rate

(FDR) respectively. We considered both direct estimates of FOR and FDR and adjusted estimates to remove mediating effects of age, sex, and SIMD. These adjusted metrics may be considered as counterfactual values after substituting one group with another (see S1 Appendix, Section 5). Note that it is possible for a risk score to be perfectly calibrated in two groups $G = g$ and $G = g'$ and still have differing FOR or FDR between groups. Moreover, in our context, false negatives are of somewhat greater concern than false positives, as they represent individuals who potentially missed out on treatment. High false positive error rates may trigger unnecessary interventions and excessive costs to the healthcare system.

Finally, to better understand the nature of false negatives, we assessed the extent to which various causes of admission using the first letter of the ICD10 code recorded as a primary admission diagnosis, (S1 Table) could be predicted in each group. Our study spans the years 2013–2018, prior to the publication of ICD-11.

For this purpose, we considered the population of individuals with SPARRA scores less than 10% ($\hat{Y} < 0.1$), which was a false negative; that is, who subsequently had an EA or died within 1 year ($Y = 1$). For individuals who died without having an EA first, we considered the ICD10 code associated with their primary cause of death. Such individuals essentially constitute an extreme example, in which an EA or death occurred against expectation.

## Ethics statement

This study and the use of NHS data was approved by the Public Benefit and Privacy Panel for Health and Social Care (study number 1718-0370; approval evidenced in application outcome minutes for 2018/19 at https://www.informationgovernance.scot.nhs.uk/pbpphsc/application-outcomes/). In addition, accessing data was approved by the Public Health Scotland National Safe Haven, through the the electronic Data Research and Innovation Service (eDRIS) and the Public Benefit and Privacy Panel (PBPP) (study number 1718-0370). All studies have been conducted in accordance with information governance standards; data had no patient identifiers available to the researchers. This work was conducted in accordance with UK data governance regulations under PBPP application number eDRIS 1718-0370.

All necessary patient/participant consent has been obtained and the appropriate institutional forms have been archived.

## Results

### Data summary

Demographic details for the individuals present in our data are shown in Table 1. Our data comprised slightly more than 50% females and slightly more than 20% individuals in the most-deprived SIMD quintile (and slightly fewer than 20% in the least-deprived quintile).

In total, 66.8% of the cohort had a recorded white ethnicity and 21.8% non-white. A substantial proportion (11.5%) of the study population had no ethnicity reported, and the absence of ethnicity data was non-random. Individuals with missing ethnicity were generally older, with higher variability of ages, potentially due to differential uptake of the COVID-19 vaccine. However, they did not include disproportionate numbers of members of the most- or least-deprived quintiles or urban or mainland postcodes (Table 1).

### Score distribution and performance

We directly analysed the distribution of SPARRA scores in each group, using cumulative distributions (also called demographic parity) and a counterfactual substitution of the alternative group. Despite sex being an input for SPARRA, there are no substantial differences in the

distribution of scores for males and females (S1 Fig, Panel B). A similar pattern was observed when comparing groups defined by urban/rural (S1 Fig, Panel E) or mainland/island (S1 Fig, Panel F) residence status. The largest difference was observed when comparing different age groups: as can be expected, individuals over 65 had much higher scores than those under 25 (S1 Fig, Panel A). The difference in score distributions between these age groups was smaller when we used a counterfactual comparison (S2 Fig, Panel A), indicating that a large part of the difference is due to between-group differences data on hospital activity and prescriptions, with the residual difference due to direct effects of age on the score (and potentially due to effects mediated through sex or SIMD distributions).

Whilst we initially observed a difference in the distribution of the scores between nonwhite and white individuals (Fig 1A), the difference largely disappeared in the counterfactual comparison (Fig 1C). Individuals in the most deprived quintile tend to have higher scores than those in the least deprived quintile (Fig 1B). This effect was somewhat reduced in the counterfactual comparison, but still present (Fig 1D). The difference in counterfactual score distributions can only be due to different distributions of age and sex in SIMD quintiles and by direct effects of SIMD on score (SIMD is an input for SPARRA). However, it is unlikely to be solely due to the former, as individuals in the most deprived quintile were of *lower* average age (whose EA risk tends to be lower), and there was little difference in sex distribution between the most and least deprived quantiles (Table 1). This indicates that the differences in counterfactual score distribution are due to the direct effects of SIMD on the risk score.

Discrimination (measured by AUROC) was generally stronger in individuals over 65 than under 25. Both white and nonwhite subgroups had poorer discrimination than the overall cohort. SPARRAv3 was generally well-calibrated across all groups. See S1 Appendix, Section 9 for details.

## False negatives and false positives

Using our hypothetical decision rule (Supplementary Note Metrics), an EA event is predicted if the estimated SPARRA score ($\hat{Y}$) is greater or equal than a score cutoff $c$. In most comparisons, we observed differences in the false positive rates between the associated subgroups, regardless of the choice of $c$. False positive rates were higher in individuals under 25 (S7 Fig, Panel A), the least deprived subgroup (S7 Fig, Panel C), nonwhite individuals (S7 Fig, Panel D), and for individuals with rural (S7 Fig, Panel E) and island postcodes (S7 Fig, Panel F) than the corresponding groups. Females and males had similar false positive rates, with females slightly higher at low thresholds S7 Fig, Panel B). All differences were virtually identical after adjusting for age, sex, and SIMD, indicating that most differences were not due to these variables (S8 Fig).

False negative rates substantially differed between all pairs of groups, across all values of $c$ (S5 Fig). For example, individuals over 65 had generally higher false negative rates than those under 25 (S5 Fig, Panel A). When considering subgroups defined by the residence postcode, we observed higher false negative rates in urban versus rural residents (Fig 2A) and for those with a mainland versus island residence (Fig 2B). These differences largely remained after adjusting for age, sex, and SIMD (S6 Fig). After adjustment, however, the gap between urban-rural groups decreased (Fig 2C), while it increased between mainland and island groups (Fig 2D). This indicated that while urban-rural differences in false negative rates were largely driven by between group differences in distributions of age, sex, and SIMD, the differences in distribution between mainland and island groups tended to cause a reduction in the difference in false-negative rate.

**Fig 1. Demographic parity measured by CDF of raw scores and counterfactual ('counterf.') scores in groups defined by ethnicity ('Eth.', panels A, C) and SIMD (panels B, D).** Counterfactual scores remove group effects on the score, which are due to different distributions of hospital activity and prescription data between groups, essentially isolating group effects to those mediated through age, sex, and SIMD. Lower sub-panels on each panel show the difference between curves. In all panels, the x-axis is in a logarithmic scale. Coloured bands show 95% confidence bands, though these are often narrow. A right-shifted distribution indicates generally higher scores.

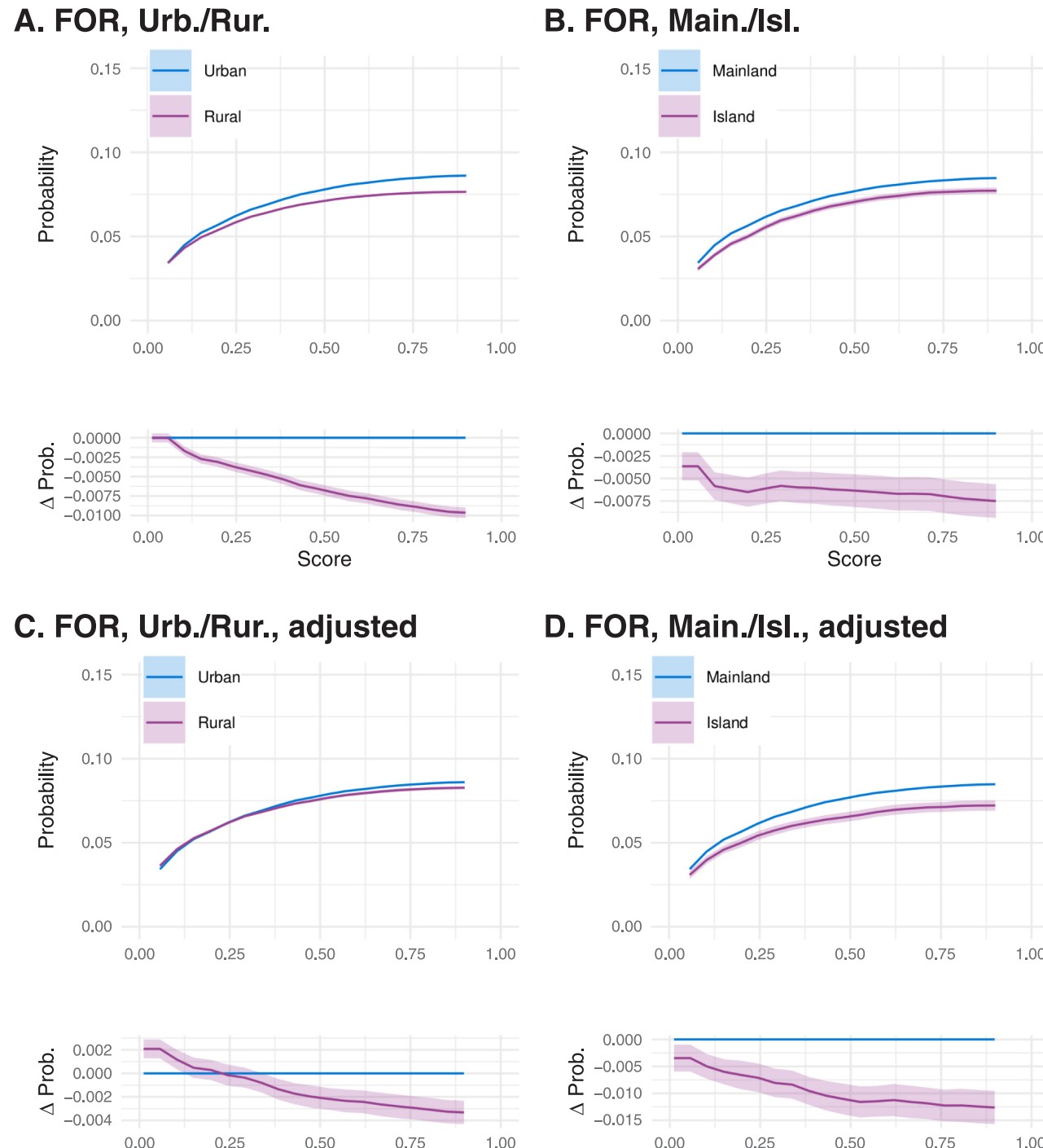

**Fig 2. FOR at a range of cutoffs for urban/rural groups ('Urb./Rur.', panels A, C) and mainland/island groups ('Main./Isl.', panels B, D).** Lower sub-panels show FOR adjusted to remove the effect of age, sex, and SIMD. Within each panel, the lower subpanel shows the difference between the two curves on the upper subpanel. Coloured bands indicate 95% pointwise confidence intervals, which are very narrow for some groups.

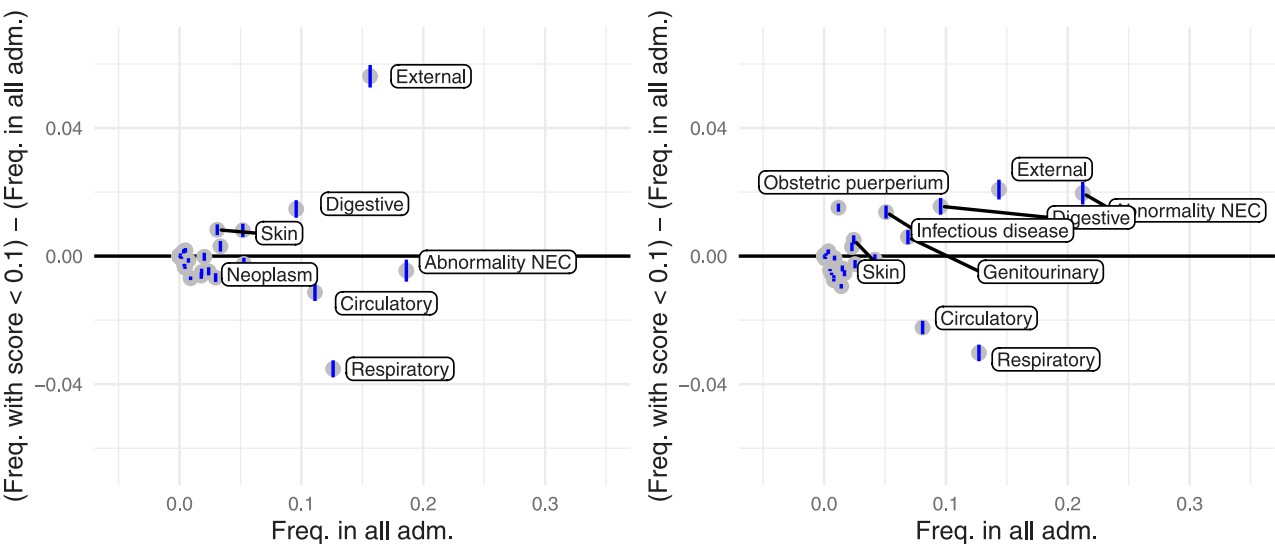

**Fig 3. Decomposition of false negatives within males and females.** Plots consider the proportion of each admission type amongst all admissions in a group (*A*) and the proportion of each admission type amongst admissions in the group with SPARRA score <10% (*B*) and plots show (*A*) against (*B*) − (*A*). Points above the line *y* = 0 correspond to admissions which are disproportionately poorly identified by SPARRA score (that is, for which $\hat{Y} < 0.1$). Blue vertical lines show pointwise 95% confidence intervals. Distinctive points are labelled.

**Decomposition of false negatives.** We explored the distribution of admission types within false negatives by considering the cohort of those with a SPARRA score less than 10% ($\hat{Y} < 0.1$) who subsequently experienced an EA or died within one year. To identify types of admission that are over- or under-represented within this cohort, we compared the frequencies of admission types in this cohort against the frequencies of admission types across all EA (regardless of the outcome predicted by SPARRA). Identifying such admission types is helpful in interpreting the scores, in that scores may be less reassuring against the risk of certain types of admissions than others.

For the overall cohort (S9 Fig), among all admissions and deaths, the most common recorded reason was due to external causes which in most cases we would not expect to be able to predict (ICD-10 codes S,T,V,X,Y; including accidents, intentional self-harm, assault and medical and surgical complications). Admissions were also frequently listed as 'abnormality NEC' (not elsewhere classified; essentially missing). Digestive, respiratory, circulatory and infectious causes of admission were common, whilst neoplastic, blood, ear, eye, and mental/behavioural admissions were rare. As expected, some admission types exhibited sex-specific patterns (e.g. obstetric admissions only recorded for females).

We then calculated the distribution of admission types restricted to individuals in each of the groups considered in this study. External causes of admission were markedly over-represented in subjects with $\hat{Y} < 0.1$ within several groups, including males (Fig 3A), individuals under 25 (S11 Fig, Panel C), individuals of white ethnicity (S11 Fig, Panel A), those in the most-deprived quintiles (S10 Fig, Panel A). The pattern was still present but to a much lesser extent in some groups, e.g. females (Fig 3B). Generally, respiratory admissions were under-represented amongst those with $\hat{Y} < 0.1$ within most groups, particularly in individuals over

65 (S11 Fig, Panel D). Circulatory admissions were over-represented in over-65 individuals with $\hat{Y} < 0.1$, but under-represented in the general population, indicating that they are relatively difficult to predict in over-65s, but relatively easy to predict in the general population (S9 Fig). Digestive causes of admission were generally disproportionately difficult to predict.

## Discussion

Our work indicates differential patterns of predictive performance for SPARRA across demographically defined groups in Scotland. For this purpose, we considered groups were defined by variables that are used as input for SPARRA (age, sex and deprivation), as well as other additional information that is not explicitly used when calculating the risk score (ethnicity and mainland/island or urban/rural residence status). The strength and direction of the differences varied across groups, and depending on the chosen performance metric. Moreover, our analysis also suggests that the interpretation of direct between-group comparisons is not always straightforward, as the observed differences may be affected by other variables, not only those used to define the groups. In combination, these findings highlight the importance of using a wide range of metrics, as well as adjusted comparisons (e.g. counterfactual) if the aim is to obtain a comprehensive characterisation for the performance of a risk score.

The SPARRA score had reasonably good calibration across all investigated groups, except for groups defined by ethnicity (for which interpretation is impeded by non-random missingness). The differences revealed by our analysis do not indicate a problem with the score itself. However, we consider that it is important for practitioners and patients to be aware of these expected differential outcomes when using SPARRA. Although our performance metrics derive from the idea of algorithmic 'fairness' between groups, we claim that we should not generally aim to change the SPARRA score so as to eliminate between-group disparities. To do so would require a change to the objective function for SPARRA away from population-level accuracy, which necessitates a sacrifice in overall performance [30, 31, 37–39]. In other words, to eliminate disparity, we would need to trade off with the score being generally less able to predict EAs. It can even be impossible to guarantee fairness constraints (in our case, interpretable as equivalence of error rates) in some cases [37], and it is often not possible to simultaneously satisfy multiple reasonable conceptions of fairness [40]. We consider that there is little to be gained from this trade-off against overall performance, in that equivalence of error rates between groups have no obvious advantage to population well-being.

The SPARRA score is well-calibrated in essentially all groups (see S1 Appendix, Section 9), indicating that it performs well at the task to which it was trained. If a practitioner aims to identify a set of individuals most likely to have emergency hospital admissions from a mixture of such groups, they should, therefore, identify the set of individuals with SPARRA scores exceeding some threshold, regardless of group status (see S1 Appendix, Section 3). Variation in error rates despite good calibration can be thought of as arising from differences in risk score distributions between groups. We argue that it is worth looking beyond calibration in this way when considering the performance of a risk score since, as discussed in the introduction, it is likely that the use of a risk score will involve decisions based on thresholds.

One of the key advantages of this study is that it uses national-level data that reflect a real-world situation and the implications of a model that has been deployed in healthcare settings for many years. Furthermore, we used multiple fairness metrics to attain comprehensive insight into the differences in scores and error rates between population subgroups. Our online dashboard enables researchers, practitioners and members of the public to explore these results interactively, providing detailed information about the performance of SPARRA.

One limitation of this study that should be considered when interpreting the results is that this study looked exclusively at routinely collected historical data, which may be subject to a variety of observational biases [41, 42]. In particular, the interpretation of ethnicity data is complex due to non-random missingness (see S1 Appendix, Section 10): our study cohort has a higher ratio of people identifying as white to people identifying as non-white than does the population of Scotland as a whole. Furthermore, our work did not consider how SPARRA is integrated into the healthcare system and did not involve the analysis of healthcare decisions (e.g. primary care interventions informed by SPARRA). As such, we are not able to identify disadvantaged groups or inequity in healthcare provision, nor recommend any changes to the distribution of healthcare resources in Scotland or to the way the SPARRA score is used. However, our findings may facilitate ongoing analyses in these areas.

Our analysis helps us understand and use the SPARRA score and provides insights into the epidemiology of emergency admissions in Scotland. In general, we demonstrate that fairness metrics are useful ways to examine patterns of errors in a risk score. We argue that analysis of this kind should be considered in the evaluation and validation of risk scores in healthcare.

## Supporting information

**S1 Appendix. Methodological details.** This appendix contains details on the SPARRA score, on metrics used in this paper, and on analytical decisions.
(PDF)

**S1 Table. Definition of different admission types.**
(PDF)

**S1 Fig. Empirical cumulative distribution of scores (log scaled) in each group, also called demographic parity.** Lower sub-panels on each panel show difference between curves. Coloured bands show pointwise 95% confidence intervals. Vertical red dashed lines identify a score of 10%. Figures for age and SIMD are replicated from the main text.
(PDF)

**S2 Fig. Cumulative distribution of counterfactual scores (log scaled) in each group.** Essentially isolating the effect of group to that mediated through age, sex, and SIMD. Lower sub-panels on each panel show difference between curves. Coloured bands show pointwise 95% confidence intervals. Vertical red dashed lines identify a score of 10%.
(PDF)

**S3 Fig. Performance by group assessed using ROC curves (discrimination).** Legends show AUROC, and associated standard errors. Lower sub-panels on each panel show difference between curves. Points on figures identify a score of 10%.
(PDF)

**S4 Fig. Performance by group assessed using calibration curves.** Lower sub-panels on each panel show difference between curves relative to whole group. Coloured bands indicate 95% pointwise confidence intervals. Points on figures identify a score of 10%.
(PDF)

**S5 Fig. False negative rates by group assessed using FOR (unadjusted); that is, $P(Y = 1|G = g, \hat{Y} < c)$ for cutoff $c$ and group $g$.** Lower sub-panels on each panel show difference between curves. Coloured bands show pointwise 95% confidence intervals. Vertical red dashed lines identify a score of 10%.
(PDF)

**S6 Fig. False negative rates by group adjusted for effect of age, sex, and SIMD, thus removing effects mediated by these.** Lower sub-panels on each panel show difference between curves. Coloured bands show pointwise 95% confidence intervals. Vertical red dashed lines identify a score of 10%.
(PDF)

**S7 Fig. False positive rates by group assessed using FDRP (unadjusted); that is, $P(Y = 0|G = g, \hat{Y} \geq c)$ for cutoff $c$ and group $g$.** Lower sub-panels on each panel show difference between curves. Coloured bands show pointwise 95% confidence intervals. Vertical red dashed lines identify a score of 10%.
(PDF)

**S8 Fig. False positive rates by group adjusted for effect of age, sex, and SIMD, thus removing effects mediated by these.** Lower sub-panels on each panel show difference between curves. Coloured bands show pointwise 95% confidence intervals. Vertical red dashed lines identify a score of 10%.
(PDF)

**S9 Fig. Decomposition of false negatives across all samples.** Plots consider the proportion of each admission type amongst all admissions in a group ($A$) and the proportion of each admission type amongst admissions in the group with SPARRA score < 10% ($B$) and plots show ($A$) against ($B$) − ($A$). Points above the line $y = 0$ correspond to admissions which are disproportionately poorly identified by SPARRA score (strictly the criterion $\hat{Y} < 0.1$)., Blue vertical lines show pointwise 95% confidence intervals. Upper plots show the proportion of (unpredicted) admissions due to each cause; lower plots show the proportion of deaths due to each cause. Distinctive points are labelled.
(PDF)

**S10 Fig. Decomposition of false negatives within SIMD and urban/rural groups.** Plots consider the proportion of each admission type amongst all admissions in a group ($A$) and the proportion of each admission type amongst admissions in the group with SPARRA score < 10% ($B$) and plots show ($A$) against ($B$) − ($A$). Points above the line $y = 0$ correspond to admissions which are disproportionately poorly identified by SPARRA score (strictly the criterion $\hat{Y} < 0.1$)., Blue vertical lines show pointwise 95% confidence intervals. Upper plots show the proportion of (unpredicted) admissions due to each cause; lower plots show the proportion of deaths due to each cause. Distinctive points are labelled.
(PDF)

**S11 Fig. Decomposition of false negatives within ethnicity and age groups.** Plots consider the proportion of each admission type amongst all admissions in a group ($A$) and the proportion of each admission type amongst admissions in the group with SPARRA score < 10% ($B$) and plots show ($A$) against ($B$) − ($A$). Points above the line $y = 0$ correspond to admissions which are disproportionately poorly identified by SPARRA score (strictly the criterion $\hat{Y} < 0.1$)., Blue vertical lines show pointwise 95% confidence intervals. Upper plots show the proportion of (unpredicted) admissions due to each cause; lower plots show the proportion of deaths due to each cause. Distinctive points are labelled.
(PDF)

**S12 Fig. Decomposition of false negatives within mainland/island groups.** Plots consider the proportion of each admission type amongst all admissions in a group ($A$) and the proportion of each admission type amongst admissions in the group with SPARRA score < 10% ($B$)

and plots show ($A$) against ($B$) − ($A$). Points above the line $y = 0$ correspond to admissions which are disproportionately poorly identified by SPARRA score (strictly the criterion $\hat{Y} < 0.1$)., Blue vertical lines show pointwise 95% confidence intervals. Upper plots show the proportion of (unpredicted) admissions due to each cause; lower plots show the proportion of deaths due to each cause. Distinctive points are labelled.
(PDF)

## Acknowledgments

The authors note that this project's success was entirely contingent on close cooperation between the Alan Turing Institute and PHS. All author contributions were significant and essential to the completion of this work. Author contributions were as follows: IT, JL, CAV, LJMA, JI, SR: have made substantial contributions to the conception and design of the work, and the acquisition, analysis, and interpretation of data. IT, JL: drafted the initial work and had full access to data. CAV, LJMA, JI, SR: critically revised it for important intellectual content. KB, RP, EL, SR, JI, JL: managed data access and disclosure. KB, JI, RP, SR: provided public health input. IT, JL, CAV, LJMA, JI, SR, RP, KB: gave final approval of the version to be published.

Computing for this project was performed on the Scottish National Safe Haven (NSH), supported by the eDRIS, and the Edinburgh Parallel Computing Centre (EPCC), based at The University of Edinburgh. The authors would like to acknowledge the support of the eDRIS Team (PHS) and the NHS National Services Scotland (NSS) for their involvement in obtaining approvals, provisioning and linking data and the use of the secure analytical platform within the National Safe Haven.

For the purpose of open access, the author has applied a Creative Commons Attribution (CC BY) licence to any Author Accepted Manuscript version arising from this submission.

## Author Contributions

**Conceptualization:** Ioanna Thoma, Simon Rogers, Jillian Ireland, Catalina A. Vallejos, Louis J. M. Aslett, James Liley.

**Data curation:** Ioanna Thoma, Simon Rogers, Jillian Ireland, Rachel Porteous, Katie Borland, Catalina A. Vallejos, James Liley.

**Formal analysis:** Ioanna Thoma, James Liley.

**Funding acquisition:** Simon Rogers, Jillian Ireland, Catalina A. Vallejos, Louis J. M. Aslett, James Liley.

**Investigation:** Ioanna Thoma, Catalina A. Vallejos, Louis J. M. Aslett, James Liley.

**Methodology:** Ioanna Thoma, Simon Rogers, Jillian Ireland, Katie Borland, Catalina A. Vallejos, Louis J. M. Aslett, James Liley.

**Project administration:** Ioanna Thoma, Jillian Ireland, Catalina A. Vallejos, Louis J. M. Aslett, James Liley.

**Resources:** Simon Rogers.

**Supervision:** Jillian Ireland, Catalina A. Vallejos, Louis J. M. Aslett, James Liley.

**Validation:** Ioanna Thoma, Simon Rogers, Louis J. M. Aslett, James Liley.

**Visualization:** Ioanna Thoma, Jillian Ireland, Katie Borland, Catalina A. Vallejos, Louis J. M. Aslett, James Liley.

**Writing – original draft:** Ioanna Thoma, James Liley.

**Writing – review & editing:** Ioanna Thoma, Simon Rogers, Jillian Ireland, Rachel Porteous, Katie Borland, Catalina A. Vallejos, Louis J. M. Aslett, James Liley.

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
